# Ambulance Use Appropriateness: Emergency Medical Service Technicians’ and Triage Nurses’ Assessments and Patients’ Perceptions

**DOI:** 10.3390/nursrep15050165

**Published:** 2025-05-09

**Authors:** Ilenia Piras, Francesca Sanna, Michele Garau, Gabriele Sole, Giampaolo Piras, Ernesto d’Aloja, Maura Galletta

**Affiliations:** 1Department of Emergency, SS Trinità Hospital, ASL Cagliari, 09100 Cagliari, Italy; ilenia_78it@yahoo.it; 2Geospatial Health and Development, Telethon Kids Institute, Nedlands, WA 6872, Australia; francesca.sanna@telethonkids.org.au; 3Medicasa Italia S.p.A., 09100 Cagliari, Italy; dgray4794@gmail.com; 4Emergency Medical Service Technician, Misericordia Association, 09100 Cagliari, Italy; 5ARNAS G Brotzu, S. Michele Hospital, 09100 Cagliari, Italy; sole.gabriele@gmail.com; 6Azienda Regionale Emergenza Urgenza Sardegna, 09100 Cagliari, Italy; mataranu@gmail.com; 7Department of Medical Sciences and Public Health, University of Cagliari, Monserrato, 09042 Cagliari, Italy; ernestodaloja@gmail.com

**Keywords:** emergency medical service, appropriateness, ambulance service, triage nurse, patient

## Abstract

**Background/Objective:** Part of the workload of ambulance service involves patients with low-acuity health events that do not require the specific resources provided by ambulance services or emergency departments (EDs). The problem of inappropriateness is also present in Italy. However, research is limited to the perspective of triage nurses only, excluding patients and emergency medical service (EMS) staff. This study aims to identify the presence of inappropriate ambulance use in the study context by comparing patients’ perception of emergency with appropriateness perspectives of both triage nurses and emergency service staff. **Methods:** A cross-sectional study involving 109 patients transported by ambulance was performed between January and March 2020. Questionnaires were distributed to patients, EMS staff, and triage nurses operating in the chief town and hinterland of a region of Italy. **Results:** Non-penetrating trauma was the most frequent cause for calling an ambulance. Patients referred that activation of the service was necessary, while triage nurses and EMS technicians were in line in believing the non-urgency of the call due to non-emergent health conditions. **Conclusions:** Although we cannot conclude that citizens use the emergency system inappropriately, the results of this study make us think about the need to implement educational interventions that increase citizens’ knowledge of how the service works and the territorial services available.

## 1. Introduction

Ambulance service workload involving patients with low-acuity health conditions that do not require the specific resources provided by ambulance services or emergency departments (EDs) is a multi-factorial problem. Responding to these cases with traditional emergency ambulance attendance and transport to a hospital ED negatively impacts ambulance service efficiency and efficacy by reducing the availability of these resources for emergency cases and thus potentially compromising patients’ outcomes [1].

The primary aim of the emergency medical service transport worldwide is to provide out-of-hospital medical care, treatment, and transportation of sick or injured people to a healthcare facility [2]. Today, emergency services have improved in the ambulance system thanks to advanced logistical technologies in operating systems and the involvement of voluntary associations collaborating with the national emergency medical system. However, the inappropriate use of emergency services by citizens is still a problem in many contexts [3,4,5]. The term “inappropriate use of emergency services” refers to the use of emergency medical services (EMSs) for non-emergency medical conditions or when the patient does not use available alternative means of transportation [6].

The efficient and effective use of EMS is crucial not only for patient outcomes but also for the effective functioning of the healthcare system as a whole. Nurses who working in emergency departments and triage settings play an important role in managing patient flow and ensuring that resources are allocated appropriately. Inappropriate ambulance use, when patients with non-urgent conditions resort to ambulance transport, can have an impact on nurses’ workload and ability to provide timely care to patients with true emergencies. This may expose professionals to an increased risk of negligence because it does not allow them enough time for meals, sleep, and training [7], thus negatively affecting their morale [8].

A pioneering study in the United States [6] investigated inappropriate ambulance use from the perspective of both EMS staff and patients who were transported to the emergency department by ambulance. Comparing participants’ perceptions, the results showed that both patients and EMS staff agreed on the necessity of ambulance transportation (54%) but also on its redundancy (20%). Moreover, 23% of participants disagreed about the perception of emergency, where EMS staff considered the use of the ambulance unnecessary. These data showed that most of the emergency transports were avoidable. In Italy, the organizational model of territorial medicine guarantees an adequate response to citizens’ health needs through the family doctor, pediatrician, and outpatient specialists during midweek daytime hours and the continuity of care physician during nighttime and holidays. However, over time, there has been a gradual increase in the number of admissions to the emergency department for patient conditions that could be adequately treated in an out-of-hospital setting without the need for the emergency system. The emergency department should not be the referral point for people with health needs that could be met in other care settings. An Italian study [9] highlighted that the problem of inappropriateness is also present in this region. In fact, the data estimated that 26.1% of the emergency transports could be avoided. However, although the study shed light on the importance of the problem of transport appropriateness in Italy, it was limited to the triage nurses’ perspective only, not considering that of patients and EMS providers/technicians. The recent literature, especially in the Italian context, lacks studies that integrate different perceptions about the appropriateness of transport. To avoid the risk of malpractice and understand how to improve ambulance use [10], it is important to compare the perspective of professionals with that of citizens/patients who make use of the service, thus investigating the degree to which the latter are aware that their medical problem is a true emergency requiring an ambulance.

Understanding perceptions of appropriateness is the first step in addressing the problem of inappropriate use. Patients’ decisions to use the ambulance service are based on their interpretation of symptoms and their perception of the need for urgent intervention. Understanding how patients assess the urgency of their requests and the appropriateness of ambulance use can provide useful information for developing effective educational interventions to increase awareness of emergency criteria and encourage more informed use of the service.

Based on this issue, the present study aims to explore and compare patients’ perception of emergency with appropriateness perspectives of both triage nurses and emergency service staff. A comparison of perceptions would make it possible to identify any discrepancies in order to identify specific patient situations or conditions that may indicate a potential inappropriate use of the service and to improve telephone triage protocols.

## 2. Materials and Methods

### 2.1. Context, Participants, and Procedure

This was a cross-sectional study carried out in a region of Italy characterized for being a large island in the Mediterranean Sea. In Italy, ambulance transportation to the hospital is provided for free by the Regional Health Service. This study was constituted by two main groups involved in the Italian Emergency/Urgency System: (1) The Operation Center under the Regional Emergency and Urgency Agency to involve EMS technicians. In Sardinia, they are unpaid volunteers, employees of social cooperatives, or employees of associations of volunteers, with basic first aid training–Basic Life Support and Defibrillation (Table A1) shows the level of assistance provided by the out-of-hospital emergency system in Italy and in Sardinia, and the personnel involved, referring to the 1996 State-Regions Agreement). (2) Triage nurses of the emergency departments of the chief town and hinterland. In the Operation Center, the triage is performed by a nurse through a structured telephone interview according to defined protocols, with the aim of assessing the request for assistance and sending the most appropriate team and vehicle to the scene. The level of patients’ criticality is identified through four codes. In the Italian emergency departments, triage nurses have specific training and use specific protocols to identify care priorities by assessing the clinical conditions of patients and the risk of progression. The assessment is based on targeted data collection, symptoms (e.g., pain), vital signs, and expected ED resource use. This allows the assignment of priority codes for access to care. At the time of this study, in Sardinia, the level of patients’ criticality was identified through four codes (Table 1). However, since 2019, in Italy, there have been five hospital triage codes: red, orange, light blue, green, and white. The yellow code (urgency) was replaced by the orange code (urgency), and the new light blue code (deferrable urgency) was introduced [11]. In Sardinia, the transition to the five codes was implemented in 2023 [12].

This study enrolled all patients who were transported by ambulance between January and March 2020 (before the beginning of restrictions due to COVID-19 lockdown). Thus, the study participants included patients, triage nurses, and EMS technicians. For patients, the inclusion criteria were agreeing to participate in this study, being at least 18 years old, and being conscious and able to communicate. Among the triage nurses were the healthcare workers on duty at the time of the patient’s transportation with the service. As for EMS technicians, on-duty volunteers were recruited, with the exclusion of trainees.

Data were collected through a structured paper check list administered to all participants after a brief explanation of the purpose of this study. The administration of the instrument to patients, EMS technicians, and triage nurses took place after patients were taken in charge by the emergency department staff and were clinically stabilized (pain-free, vital signs stabilized, etc.). Before completion, a unique sequential code was assigned to each check list to match different check lists referring to the same patient. The completed check lists were archived in locked folders under the direct control of the researchers.

### 2.2. Ethical Statement

Participation in this study was voluntary and anonymous for all participants, and it was subject to the signature of the informed consent form. Authorizations to recruit participants were requested by the Operation Center, the Regional Emergency and Urgency Agency, and both the health directors and the head nurses of the emergency departments participating in this study. This research was conducted in accordance with the 1975 Declaration of Helsinki in its current, revised version. The approval by the Ethics Committee was not needed because this research did not include patients’ sensitive information but just aggregated perception data. In Italy, no ethical approval is required based on the 211/2003 Italian law and the General Data Protection Regulation (EU) 679/2016 (“GDPR”) provisions for observational studies as they are not considered clinical research. However, all participants were informed about the purpose of this research and provided with their written informed consent.

### 2.3. Instruments

A structured paper check list was administered to patients, triage nurses, and EMS technicians. The instrument was structured based on the available literature on the topic [6] and adapted according to the contextual needs and the purpose of this study. The check list was tested using a pilot sample of two citizens, two nurses, and two EMS technicians to ensure the clarity, suitability, and comprehensibility of the instrument.

#### 2.3.1. Patient Check List

The check list included ten items. Three items regarded demographic information such as gender, age, and education. The other items included dichotomous answers (yes, no) and response options (to be chosen from the listed options) regarding possession of a personal car and/or availability of a family network to reach the emergency department, the reason for the call, the time frame in which it occurred, and whether the patient had already used the ambulance service before. At the end of the instrument, there was a section for the patients to leave their opinion on whether or not an ambulance with a patient on board takes precedence over patients arriving at the emergency department by their own means.

#### 2.3.2. Triage Nurse and EMS Technician Check List

The check list for both triage nurses and EMS technicians consisted of eight items. Three items collected demographic information (age, gender, length of service). The other items included dichotomous answers (yes, no) and response options (to be chosen from the listed options) regarding the need for ambulance transport for the patient by indicating why it was deemed appropriate or not. The last item regarded the main cause or medical problem for which the patient called an ambulance. These were based on the triage and clinical assessment tools used by professionals to determine the reasons for ambulance calls and emergency department arrivals. In the instrument dedicated to EMS technicians, the lead EMS clinician was responsible for indicating the alphanumeric code of the ambulance dispatch so that it could be compared with any code change made by the emergency department following the triage process. The instrument for nurses included two items regarding whether the color code was changed from the one assigned by the Operation Center and the details of the code change.

EMS technicians expressed their opinion about the appropriateness of ambulance use on the basis of the vital parameters and signs detected in patients, as well as in relation to the indications received from the nurse dispatcher of the Operation Center. In this sense, their evaluation was objective and supported by the methods acquired during specific training. Triage nurses expressed their opinion about the appropriateness of ambulance use based on triage decision-making standardized protocols. This approach included objective elements such as patients’ medical history, clinical signs, symptoms, and vital signs for patients who arrived in the emergency department. After that, the nurse assigned a priority class for the timing of access to the medical examination.

### 2.4. Analysis

Data analysis was carried out using IBM SPSS software 20.0. Frequency and percentage were estimated to analyze the sample characteristics, and the chi-square test (χ^2^) was used to analyze any statistically significant differences (*p* < 0.05) in the perception of ambulance use among the three groups of participants (patients, triage nurses, and EMS technicians) and the reasons why the ambulance was not necessary, according to nurses’ and EMS staff’s assessment.

## 3. Results

The sample included 109 out of 156 (response rate: 69.9%) patients who called an ambulance and met the inclusion/exclusion criteria of this study. Forty-seven patients were excluded because they did not meet the inclusion criteria. To assess potential selection bias, we compared the patients included in this study (N = 109) with the patients excluded (n = 47) in terms of gender and age. The chi-square analysis showed no significant differences in the proportion of male and female between the two groups (χ^2^(1) = 0.124, *p* = 0.725). Similarly, there was no statistically significant difference in age range between the included and excluded patients (χ^2^(4) = 9.320, *p* = 0.054). These results suggest that the included sample does not differ significantly from the excluded group for the studied demographic variables, thus reducing the likelihood of selection bias based on gender and age.

In addition, we assessed whether our sample of 109 patients was representative of the general population in terms of gender, age, and level of education, using data from the National Statistics Institute (ISTAT) available for the year 2021. The results showed that the sample was representative of the general population in terms of gender (χ^2^(1) = 2.543, *p* = 0.111) but not in terms of age (χ^2^(4) = 19.594, *p* < 0.001) and education (χ^2^(2) = 19.789, *p* < 0.001). As the age range sets for our sample were slightly different from those in the census for the general population, the two groups may not be comparable, so we decided to apply weighting by educational level. Before weighting, the sample showed an over-representation of individuals with a secondary (high school) education level (52.3% in the sample vs. 31.0% in the population) and an under-representation of individuals with an elementary or middle school education level (20.2% in the sample vs. 34.5% in the population). Although the age range in the general population does not overlap with those defined for our sample, the distribution showed some deviations from the population, with an over-representation of the 31–40 years age range (sample 13.8% vs. population 10.4%), 41–50 years age range (sample 20.2% vs. population 14.5%), 51–60 years age range (sample 19.3% vs. population 16.9%), and an under-representation for the >60 years age range (sample 26.6% vs. population 36.6%). The application of the weights changed the distribution of the sample, bringing it closer to the population data. After weighting, the proportion of individuals with a secondary (high school) education in the sample decreased to 43%, while the proportion of individuals with an elementary or middle school education increased to 34%. The age distribution, although still showing deviation from the population for the age groups 18–30 years (sample 20.6% vs. population 12.3%) and 41–50 years (sample 19.2% vs. population 14.5%), also came closer to the population distribution, reducing the over- and under-representations observed in the unweighted sample. The 31–40 age range decreased from 13.8% to 12.2%, the 51–60 age range decreased from 19.3% to 17.7%, and the 61–70 age range increased from 26.6% to 30.5%.

### 3.1. Demographics

The following analyses were conducted using sample weights, resulting in a weighted effective sample size of 131 patients, which was larger than the observed sample size of 109 patients. The majority of the sample is represented by men (n = 71, 54.2%; women n = 60, 45.8%; other n = 0, 0%) who called an ambulance. About 21% (n = 27) of the patients were aged between 18 and 30, and 19% of patients were between 41 and 50 years. Patients aged 51–60 years were 17.6% (n = 23), followed by patients aged 31–40 years (12.2%, n = 16). Patients aged > 60 years accounted for the highest percentage (30.5%, n = 40) of patients who called an ambulance. Most patients had a secondary education (high school) (n = 57, 43.5%), followed by patients with both primary (elementary and middle school) (n = 44, 33.6%) or academic (bachelor’s or master’s degree) education (n = 30, 22.9%).

### 3.2. Sample Characteristics

The most frequent problem for which the patients activated an ambulance was non-penetrating trauma such as fractures associated with traffic accidents (26.0%, n = 34) (Figure 1).

The majority of the participants indicated that the ambulance activation was necessary (n = 104, 79.4%), and 20.6% (n = 27) referred that the transport was not necessary, but they were urged to call the emergency service by a family member. Moreover, most of the respondents (n = 96, 73.3%) reported that they used the ambulance service for the first time in their lives, while 26.7% of participants (n = 35) had previously activated the ambulance service due to an acute disease (46%, n = 16), chronic disease complications (37%, n = 13), or trauma or injuries (17%, n = 6). A total of 55 participants (42.0%) said that they activated the ambulance service immediately or a few minutes later from the onset of the issue, while lower percentages were found for those who said that they activated the service within 30 min (n = 24, 18.4%) after one hour but within two hours (n = 19, 14.5%), after two hours or more (n = 33, 25.%). Regarding the priority of ambulance-transported patients over those arriving at the emergency department by their own means, most of the participants (n = 68, 51.9%) referred to the fact that patients transported by ambulance had priority over others. The distribution of the data by age group showed that the percentage of patients according to whom ambulance transport provides priority access to care increases with age (χ^2^(4) = 15.1, *p* < 0.01) (Table 2).

### 3.3. Viewpoints of Patients, Triage Nurses, and EMS Technicians

Patients’ perceptions of the need to activate the ambulance service differed significantly from those of triage nurses and EMS staff (χ^2^ = 27.805, *p* < 0.001). Specifically, while 79.4% of patients believed the ambulance necessary, both triage nurses and EMS staff agreed that, for a significant percentage of patients (47% and 49%, respectively), activating the service was not necessary (Table 3).

Most patients (66.3%, n = 69) motivated the need for an ambulance by claiming their inability to reach the emergency department because of other symptoms related to the episode or chronic disease. Other reasons provided were the lack of personal means of transportation (15.4, n = 16) or social networking (friends and/or relatives with personal means of transportation) (7.7%, n = 8), or because the emergency department was too far away (10.6%, n = 11).

The chi-square test indicates that there were no significant differences between the distributions of the reasons given by triage nurses and EMS technicians (χ^2^ = 0.413, *p* > 0.05) for not requiring an ambulance, suggesting concordance in their assessments. The main reason given by both the groups as a result of their assessment of the patient concerned minor health problems that did not require transport to the emergency department by ambulance (35.5% of patient for nurses, 40.6% of patients for EMS staff) (Table 4).

Specifically, triage nurses indicated that 62 out of 131 patients did not need an ambulance. Figure 2 shows that the main problems of the patients transported to the emergency department were related to other symptoms and disorders (e.g., psychomotor agitation state, dermatological symptoms or complaints, social problems) considered non-urgent (30.6%), followed by abdominal pain (16.0%) and non-penetrating trauma (14.5%).

Regarding the demographic characteristics of the patients who did not need an ambulance intervention based on the nurses’ assessment (n = 62), it was found that 35.5% of them were more than 60 years old, followed by patients aged 18–30 years (22.6%) and by patients aged 41–50 years (19.4%). The majority of the sample (n = 32, 51.6%) had a medium or high level of education (secondary school or university) and referred to the need to activate the service (69.4%) because of their inability to reach the emergency department due to other symptoms related to the episode or chronic disease (58.1%) (Table 5).

The priority codes assigned to the patients by the operation center of the emergency service were green (low criticality) at 45.0% (n = 59), yellow (moderate criticality) at 53.4% (n = 70), and red (high criticality) for two cases only (1.5%). There was no evidence for white code (no criticality).

Among them, 22 out of 113 (16.8%) codes were changed after the assessment by the nurse following the triage access (Table 6). Specifically, 11 codes (50%) changed from green to yellow and 11 (50%) from yellow to green.

## 4. Discussion

The results of this study contribute to a shared understanding of the perceived appropriateness of ambulance use. These findings show a higher prevalence of men who activated the ambulance service than women, and this is in line with the pioneering study by Richards and Ferrall [6], who also observed a higher rate of ambulance use among men. This would suggest that gender may play a role in the use of emergency services. Adults aged 41-50 and 18-30 were the groups that most frequently activated the ambulance service for the first time. Among them, most people who activated the ambulance service had a medium or high scholastic level. This suggests that the service activation is not related to people’s educational level. In fact, the reason most frequently reported by patients is the inability to reach the emergency department independently due to other symptoms related to the episode or chronic disease. This study also shows that patients who, in the opinion of nurses and EMS, did not need to be transported by ambulance were mainly elderly (>60 years) with medium-high educational qualifications and felt that ambulance use was necessary because of other symptoms related to the episode that prevented them from reaching the emergency department independently. This underlines the need to develop care pathways for chronic patients to avoid relapses and repeated hospitalizations.

About one-third of the study participants reported that the use of the ambulance was not necessary and that they were prompted to call the service by family members or friends, or the family members or friends themselves called it on their own initiative. This may be because understanding the health status of others is not as easy as understanding one’s own needs. In this sense, the decision to call an ambulance is based on an emotional response related to perceived priority and urgency, leading the family members to choose the path of least risk, which is to call an ambulance. This is also highlighted by previous research showing that relatives can help the patient to realize that sometimes there are no alternatives and that the ambulance is necessary [13]. In addition, the results of this study indicate a higher turnout of males who called an ambulance than females. In effect, in line with the previous research [6,14], the highest percentage of problems for patients who came to the emergency department by ambulance involved non-penetrating trauma as a result of a road accident. Since the incidence of traffic accidents in males of all ages is higher than in females [15], this finding may explain the high frequency of males who activated the service. Another interesting finding is that a high percentage of patients believe that the arrival of an ambulance guarantees priority treatment. As a result, they may be led to believe that the arrival of an ambulance means a more critical condition and therefore priority treatment, thus reducing waiting time. This also emerged from previous research [16,17,18] and highlights the opportunity for nurses to educate patients about the triage process and how patients are prioritized according to the severity of their condition, not how they arrived. However, about 73% of patients referred to have activated the emergency system for the first time in their lives. This suggests that, for most participants, calling an ambulance is not a routine event but a response to a unique event assessed as truly urgent. This research showed that first-time callers may, on the one hand, be uninformed about the available alternatives, and, on the other hand, the decision to activate the service is made when they feel that they have no other choice [6,13].

Both EMS technicians and triage nurses reported that about 50% of patients did not need the ambulance service based on their health condition. This finding highlights a specific decision-making process for nurses and EMS, based on their professional responsibility and training, such that they evaluate (and face to) the same problem differently from patients and their families. Approximately 30% of patients reported to be aware of their non-critical health condition although they called the ambulance anyway. It is possible that these patients have had difficulty quickly accessing other forms of healthcare (primary care physician, doctor on call, etc.) and see the ambulance as the fastest way to receive medical care.

Most of the priority codes for patients arriving in the ED were yellow, likely because the patients’ conditions were described as urgent. However, the results also showed a high percentage of green priority codes. An explanation for this finding may be that, although the patient’s condition did not warrant activation of the service, according to the nurses and EMS, the responsibility for the decision rests with the nurse in the Operation Center. In this sense, as supported from previous research, the choice to send an ambulance to the patient is considered by the nurse to be the most cautious one [19,20,21]. It is important to notice that only a small proportion (less than one-fifth) of the priority codes were downgraded or upgraded after the triage nurse’s assessment. This consistency between the initial telephone assessment and the nursing assessment in the presence of the patient indicates the reliability of the initial telephone triage and thus the effectiveness of the protocols used in distinguishing situations of different severity. In addition, among the priority codes that were changed in color based on the nurse’s assessment, half of them went from yellow to green. Although we cannot objectively claim that the patient’s condition was not critical because our study did not conduct a clinical evaluation, this finding would suggest that some patients could be managed by local medical services available in the community, such as primary care physicians, the continuity of care services, and local health and social services. There may be a lack of awareness among citizens or a lack of integration between the different levels of care (emergency/urgency and primary/territorial care), leading to the inefficient use of resources. It is critical to invest in and strengthen territorial services to make them more accessible, efficient, and capable of handling a wide range of non-emergency conditions.

Although we cannot conclude that citizens use the emergency system inappropriately, the results of this study make us think about the need to implement educational interventions that increase citizens’ knowledge of how the service works and the territorial services available.

### 4.1. Limitations and Strengths

This study has limitations. First, there is no objective evaluation criterion to compare patients’ opinion with both nurses’ and EMS’ evaluations. Patients’ opinions about the appropriateness of using an ambulance were based on their experience and perceived need for intervention. However, patient-facing objective assessment tools measuring the appropriateness of ambulance use are not available in the literature. Thus, it was not possible to evaluate the discrepancies of opinion among different participants using objective criteria for all three categories. However, the perceived appropriateness of the ambulance use by both nurses and EMS was assessed through objective elements, such as service checklists, protocols, and guidelines.

Second, the use of color codes as a measure of appropriateness may be limiting. Patient decision to use emergency services is based on their personal interpretation of symptoms and level of concern, which may not be consistent with objective clinical severity. Considering that we did not conduct an assessment of clinical appropriateness—which would require a review of medical records and diagnostic results—future research should integrate objective clinical data to compare perceived appropriateness with clinical appropriateness, thus providing a more complete understanding of the phenomenon.

Third, a concern that could result in a limitation of this study is that the research was interrupted due to the onset of the SARS-CoV-2 pandemic and lockdown restrictions that prevented us from collecting more data. Data collection lasted three months, but it was planned for a six-month period. This resulted in a smaller sample size and a shorter than expected data collection period, which may have led to potential imbalances in the sample. However, we assessed possible imbalances for demographic variables with respect to the general population and applied weighting techniques to correct the imbalances. In this way, the distribution of the sample was more closely aligned with that of the population, thus improving the representativeness of the sample and the generalizability of the results.

Finally, this research was conducted within a specific Italian region, and this may limit the applicability of its results to other regions within the same country. However, the literature reveals a geographical variability in ambulance use, influenced by health status, the socio-economic status of people, and the local availability of health services [22]. This suggests the need to use targeted strategies according to regional characteristics to optimize the effectiveness of the ambulance service in specific contexts. In addition, it is also reported in the literature that, in a considerable number of cases, the call to the pre-hospital emergency system does not result in transport of the patient to the hospital, indicating that, in many interventions, the rescued persons had no need for treatment [23]. Regional health policies could probably influence the decision to call for ambulance services by the population or the choice to consider other available services. An important strength of this study is the use of different perspectives. Therefore, assessing the phenomenon from the perspective of patients, EMS technicians, and triage nurses allows us to limit the biases generated by a single information source, thus improving the quality of the results [24].

### 4.2. Relevance to Nursing

The results highlight the importance of accurate triage and assessment by nurses both in the pre-hospital setting (e.g., telephone triage) and in the emergency department. The discrepancy between patient perceptions and professional assessments highlights the need for standardized protocols and ongoing training for nurses in the recognition and management of non-urgent conditions. Nurses are often the first point of contact for patients seeking emergency care, and their ability to effectively triage and refer patients to the most appropriate level of care is critical to optimizing resources. Developing and implementing patient education programs, both in the community and within healthcare facilities, is important. Nurses can play a pivotal role in planning these programs, adapting them to specific populations and evaluating their effectiveness.

### 4.3. Implications for Emergency and Interventions

Ambulance use inappropriately can increase emergency nurses’ workload and affect the quality of the services provided by emergency departments. Policies and means are needed to offload the emergency service from avoidable calls and allow emergency nurses to be able to take better care of patients in need of true urgent care. Additional methods, such as a secondary telephone triage for emergency patients, could reduce unnecessary ambulance use [1,25] and improve the quality of services. The results would suggest a need to educate citizens about the functioning of triage systems in emergency departments. Targeted educational campaigns—utilizing informational pamphlets and brochures distributed in various settings such as emergency departments, community health services, and pharmacies—should be used to support patient education about the appropriate use of ambulance services and raise awareness on the existence of alternative medical services to manage problems related to patients’ chronic or social conditions. To improve public understanding of appropriate ambulance use and triage procedures, local television, radio, and social media could be used to disseminate information explaining how triage nurses assess patients and assign priority codes based on the severity of conditions, not the mode of arrival. Policy changes could consider to develop triage and ambulance dispatch protocols that clearly define criteria for the use of alternative services in non-emergency cases to optimize the use of emergency resources. In addition, educational actions by nurses could also be aimed at younger people through training initiatives in schools that sensitize younger people to disseminate the information they learnt to their families as well. This can be an effective way to educate the community, along with the involvement of centers for the elderly and support groups.

Knowing the available health services, how to access them, and how to use them properly can increase citizens’ responsibility for the proper use of the services, thus improving their effectiveness. Future research could focus on developing and testing nurse-led interventions, such as telephone triage protocols or community-based education programs, to analyze their effectiveness in empowering citizens, making them more aware and able to recognize their own needs for care and identify the most appropriate facility to respond to them.

## 5. Conclusions

Potential inappropriate ambulance use by general population is a problem that can affect the quality of services provided by emergency departments because it may increase workload for staff and adversely affect the quality of care for patients with true emergencies. Policy strategies should include, on the one hand, targeted campaigns to educate citizens about the appropriate use of ambulance services and the availability of alternative healthcare resources. On the other hand, policy interventions should strengthen primary care, the continuity of care, and local health and social services to provide viable alternatives to ambulance use for non-emergency conditions and more effectively manage chronicity.

The results of this study suggest that this phenomenon could be present in the studied context and needs to be deepened and addressed in future studies.

## Figures and Tables

**Figure 1 nursrep-15-00165-f001:**
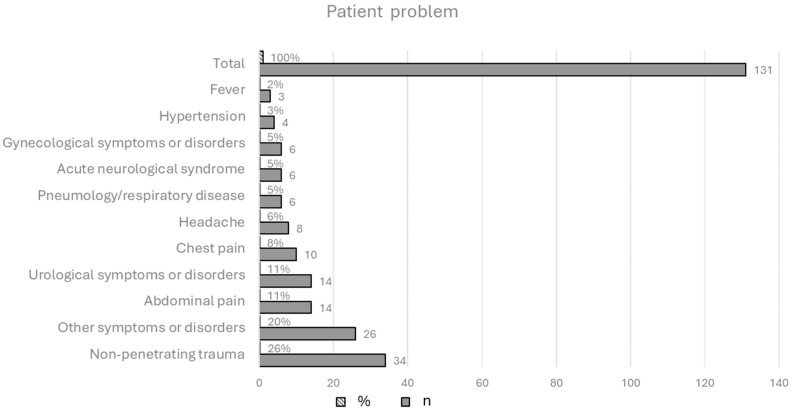
Patient problem for which an ambulance was called; N = 131 (weighted sample).

**Figure 2 nursrep-15-00165-f002:**
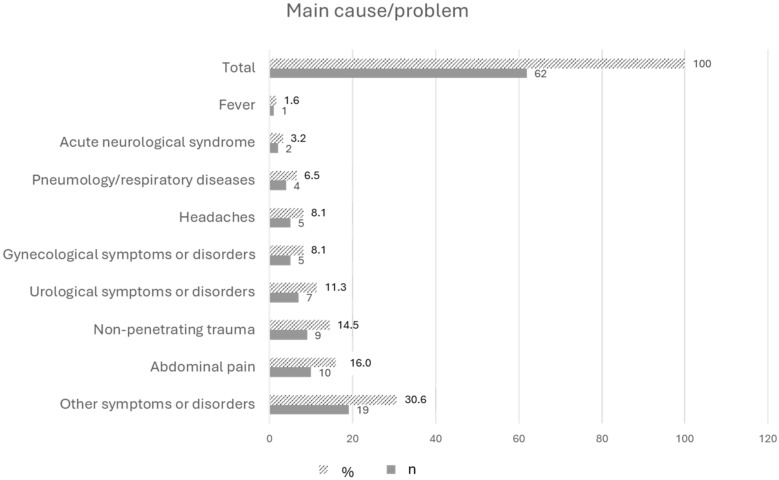
Main problems of the patients who did not need an ambulance; n = 62 (weighted sample).

**Table 1 nursrep-15-00165-t001:** Triage color coding and definition of patients’ priority characteristics.

	Operation Center	Emergency Department
Code	Priority	
Red (Emergency)	Emergency intervention on patients with impairment of more than one vital function	Compromised vital functions, immediate access to care
Yellow (Urgency)	Undifferentiated urgent intervention on patients with impairment of at least one vital function	Condition with risk of evolution or intense pain
Green (Minor urgency)	Deferrable intervention	Stable condition with no evolution risk that usually requires simple, single-specialty therapeutic services
White (No urgency)	Service that does not need to be performed quickly with reasonable certainty	Non-urgent problem of minimal clinical relevance

**Table 2 nursrep-15-00165-t002:** Age distribution of patients about perceived priority of care for people transported by ambulance.

Age Range	Do Patients Transported by Ambulance Take Priority Over Patients Who Came to the Emergency Department Independently?			
	Yes		No	
	n	%	n	%
18–30 years (n = 24)	8	37.3	13	62.7
31–40 years (n = 15)	6	45.1	8	54.9
41–50 years (n = 22)	6	26.4	16	73.6
51–60 years (n = 21)	11	60.4	7	39.6
>60 years (n = 15)	24	74.8	8	25.2

Note: N = 131 (weighted sample).

**Table 3 nursrep-15-00165-t003:** Participants’ view about the need for an ambulance.

Need for an Ambulance	Patients n (%)	Triage Nurses n (%)	EMS Technicians n (%)	Chi-Square (χ^2^)
Yes	104 (79.4)	69 (52.7)	67 (51.1)	27.805, *p* < 0.001
No	27 (20.6)	62 (47.3)	64 (48.9)	
Total	131 (100)	131 (100)	131 (100)	

Note: N = 131 (weighted sample).

**Table 4 nursrep-15-00165-t004:** Reasons why the ambulance was not necessary, according to nurses and EMS staff.

No Need for an Ambulance (Why?)	Triage Nurses Patient n (%)	EMS Technicians Patient n (%)	Chi-Square (χ^2^)
- Disease requiring medical intervention but not ambulance transport to the emergency department	21 (33.9)	21 (32.8)	0.413, *p* = 0.813
- Minor disease, not requiring transport to the emergency department	22 (35.5)	26 (40.6)	
- Non-urgent health problem	19 (30.6)	17 (26.6)	
- Total	62/131 *	64/131 *	

Note: * weighted sample.

**Table 5 nursrep-15-00165-t005:** Characteristics of the patients who did not need an ambulance according to the triage nurses’ assessment.

Patient Characteristics		Patients Who Did Not Need an Ambulance Based on the Nurses’ Assessment (n = 62 *)	
		n	%
Age	18–30 years	14	22.6
	31–40 years	4	6.5
	41–50 years	12	19.4
	51–60 years	10	16.1
	>60 years	22	35.5
Education	Low level (elementary and middle school)	30	48.4
	Medium level (secondary school)	21	33.9
	High level (university)	11	17.7
Patients’ perceived need for an ambulance	Yes	43	69.4
	No	19	30.6
If yes, why?	Inability to reach the emergency department due to other symptoms related to the episode or chronic disease	25	58.1
	Lack of own means of transportation	4	9.3
	Lack of a social networking (friends and/or relatives with means of transportation)	6	14.0
	Emergency department too far away	8	18.6

Note: * weighted sample.

**Table 6 nursrep-15-00165-t006:** Percentage of the color code assigned by the operation center and changed after the nurses’ evaluation.

Codes Assigned by the Operation Center of the Emergency Service N (%)	Changed Codes After the Nurse Assessment n (%)	Unchanged Codes After the Nurse Assessment n (%)
131 * (100)	22 (16.8)	109 (83.4)
	11 from green to yellow	
	11 from yellow to green	

Note: * weighted sample.

## Data Availability

Due to privacy and ethical restrictions, the data are available upon request to the corresponding author.

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
