# Peer review of "Ambulance Use Appropriateness: Emergency Medical Service Technicians’ and Triage Nurses’ Assessments and Patients’ Perceptions"

_nursrep, 2025, doi:10.3390/nursrep15050165_

Round 1
Reviewer 1 Report
Comments and Suggestions for Authors
Please see the attached file for my comments.

Author Response
Reviewer 1
Legend: Comment, Response
The manuscript “Ambulance use appropriateness: emergency medical service technicians’ and triage nurses’ assessments and patients’ perceptions” examines the appropriateness of ambulance use by comparing patient perceptions with assessments from triage nurses and emergency medical service (EMS) technicians, highlighting discrepancies and the need for public education on emergency system utilization. The problem addressed is both interesting and significant, and the findings will likely be of interest to others in this field. To further strengthen the manuscript, I encourage the authors to consider the following comments (listed in order of importance):
- The introduction outlines the issue of ambulance misuse, but it lacks a strong justification for why patient perceptions are being compared to EMS and triage nurse evaluations. It would be helpful to explicitly state how this comparison contributes to improving ambulance service efficiency or patient education.
1. Thanks for the suggestion. We integrated this aspect in the introduction (page 2) as requested to provide more clarity.
- While the study presents statistical comparisons among patient, EMS, and triage nurse perceptions, there is limited discussion on the statistical significance of the results. It would strengthen the manuscript to include more details on how statistical tests were used and their implications on the findings. I suggest the authors provide more in-depth statistical analysis and clearly discuss the significance of the results.
2. Thank you for your comment. We integrated the analysis section (2.4) and rewritten the part of the results better explaining statistical significance (Section 3.3, Pages 5-6).
- The discussion touches on patient education and triage system awareness as potential interventions, but it does not explore in detail how these could be practically implemented. A clearer roadmap for future interventions or policies based on the findings would enhance the study’s impact. I suggest the authors expand on the implications of the findings, offering concrete recommendations for intervention or policy changes.
3. Thank you for the comment. We agree that providing more details for future interventions and policy would strengthen the manuscript. We expanded the discussion (section 4.3) by offering concrete suggestions.
- The study is conducted within a specific region in Italy, which may limit the applicability of its findings to broader populations. A discussion on how the results compare to international studies or how regional healthcare policies influence ambulance use would help generalize the findings for a wider audience.
4. Thank you for your comment. We discussed this issue as requested in the limitation section (Page 11, Section 4.1, fourth paragraph).
Reviewer 2 Report
Comments and Suggestions for Authors
This manuscript analyses the issue of inappropriate ambulance use by the Emergency Medical Service (EMS), which has significant implications for healthcare resource allocation. Furthermore, its comparative analysis of the perceptions of multiple stakeholders - patients, triage nurses and paramedics - represents a valuable approach. In terms of study design and methodology, this is a well-structured cross-sectional study involving 109 participants, including patients, triage nurses and paramedics. The results and discussion outline the discrepancy between patients' perceptions of emergency situations and the professional assessment of paramedics and triage nurses. Practical solutions, such as educating the public about the correct use of EMS, are suggested as alternatives that will benefit all readers in the profession.
Author Response
We would like to thank the reviewer for his/her words of appreciation of our work and we are happy that our research was appreciated.
Reviewer 3 Report
Comments and Suggestions for Authors
I have read with interest the work from Piras et al., who investigate EMS users perceptions of the service. While the work address an important topic, I believe there are several issues to be addressed, and is not suitable for publication in its current form.
Introduction
1) The study aim is incoherent in current formulation. The authors state the aim is to "identify the presence of inappropriate ambulance use", but then they move to analyze the users and staff perceptions of appropriateness, which may differ from the actual clinical condition. I.e. a patient with "toothache" could be induced by his relatives to call for EMS for a problem he percieves as non-urgent (=inappropriate), but it could actually be a hearth attack symptom, and therefore EMS recourse was, in fact, appropriate. The rest of the paper correctly analyzes percieved appropriateness, but the issue of clinical appropriateness is never actually brought up anywhere in the study.
Similarly, usage of color codes as appropriateness proxy is problematic, as a "healthy" patient may not have the base knowledge to recognize a symptom as trivial, and therefore, for his own point of view, recourse to EMS is appropriate since only a health professional may recognize his problem as clinically irrelevant.
Methods:
2) The Authors' describe a local regional system, where EMS personnel is comprised by "EMS technicians (in Italy, they are volunteers with basic first aid training–Basic Life Support and
Defibrillation)". This description, in its current form, implies that:
- There is no advanced EMS ambulance in the Region (ambulance with professional nurse of physician);
- All Italian regions follow the same model.
While I am not sure how sardinian personnel is structured in EMS provision, I am positive that several Italian regions have different staffing rules for EMS; for instance, all Ambulances in the Lazio region are staffed by a professional nurse. In other regions, a quota of ambulances are staffed with medical personnel as well.
The Authors should clarify these two points, with appropriate references.
3) In Italy, pre-hospital triage works on 4 codes as described in Table 1. However, since 2019, Emergency departments codify their triage on a 5-code basis. The two codes are not equivalent, as one is a standard for pre-hospital triage, while the other is a standard for hospital triage. Did the sardinian system differ from the national standards at the time of the study? The Authors should increase clarity on this point.
4) The study only included patients who were willing to participate. It is unclear if the resulting sample is representative of the whole patients population (e.g., age distribution, gender ratio, socioeconomic factors, clinical condition, clinical relevance). This could lead to selection bias, limiting external validity. Moreover, the sample may not be representative of the general population, again limiting the study validity. Formal statistical tests should be performed to verify representativeness (es. goodness-of-fit test comparing sample demographics to patients demographics, and also to census data). If the sample is not representative, adjustments should be made (es. weighting factors) to correct for demographic imbalances.
5) The study mentions usage of two different quesitonnaires, one for patients, one for healthcare professionals. However, there is no real information on those questionnaires. What is their origin? were they validated? what are their reliability metrics?
6) "The study also enrolled all patients who were transported by ambulance between 110
January and March 2020 ". Does "also" means there is another cohort enrolled in a different period?
Results
7) The results section may need rework, as per eventual methods adjustments suggested. In particular, the statistical analysis seems mainly focused on descriptive aspects, and not accounting for potential sampling imbalances. Inferential elements are reduced to a single chi-squared test.
8) No detail is provided to questionnaire items response.
Discussion
9) Results should be addressed in the context of current literature, instad of just summarizing the findings.
10) Methodological limitation of the study are not taken into account in discussions. There is no deep evaluaiton of such limitation, nor alternative interpretations of the findings are provided.
11) There is no indication of possible generalization of the findings.
Conclusions
12) Authors should consider to add insights on clinical relevance and policy implications of their findings.
Author Response
Reviewer 3
Legend: Comment, Response
I have read with interest the work from Piras et al., who investigate EMS users perceptions of the service. While the work address an important topic, I believe there are several issues to be addressed, and is not suitable for publication in its current form.
Introduction
1) The study aim is incoherent in current formulation. The authors state the aim is to "identify the presence of inappropriate ambulance use", but then they move to analyze the users and staff perceptions of appropriateness, which may differ from the actual clinical condition. I.e. a patient with "toothache" could be induced by his relatives to call for EMS for a problem he percieves as non-urgent (=inappropriate), but it could actually be a hearth attack symptom, and therefore EMS recourse was, in fact, appropriate. The rest of the paper correctly analyzes percieved appropriateness, but the issue of clinical appropriateness is never actually brought up anywhere in the study.
1) Thank you for pointing this out to us. We clarified the aim to be more consistent with the results (Page 2). Based on Reviewer 1's suggestion, we also clarified why it is useful to compare the perceptions of professionals and patients. Regarding the clinical appropriateness issue, thank you for this observation. You are right, we did not carry out an assessment of clinical appropriateness and did not mention it in the manuscript. We think it is important to point this out and we mentioned the issue in the limitation section (Page 11, second limitation).
Similarly, usage of color codes as appropriateness proxy is problematic, as a "healthy" patient may not have the base knowledge to recognize a symptom as trivial, and therefore, for his own point of view, recourse to EMS is appropriate since only a health professional may recognize his problem as clinically irrelevant.
Thank you. We addressed this aspect in the limitations, suggesting the importance for future studies to integrate objective data on clinical appropriateness (second limitation, Page 11).
Methods:
2) The Authors' describe a local regional system, where EMS personnel is comprised by "EMS technicians (in Italy, they are volunteers with basic first aid training–Basic Life Support and
Defibrillation)". This description, in its current form, implies that:
- There is no advanced EMS ambulance in the Region (ambulance with professional nurse of physician);
- All Italian regions follow the same model.
While I am not sure how sardinian personnel is structured in EMS provision, I am positive that several Italian regions have different staffing rules for EMS; for instance, all Ambulances in the Lazio region are staffed by a professional nurse. In other regions, a quota of ambulances are staffed with medical personnel as well.
The Authors should clarify these two points, with appropriate references.
2) Thank you for your comment. We specified in the paper (Section 2.1) that in Sardinia EMS technicians are unpaid volunteers, employees of social cooperatives or employees of associations of volunteers. We have also added Attachment 1 with a table showing the levels of out-of-hospital emergency care in Italy and in Sardinia, including the type of personnel involved (Attachment 1).
3) In Italy, pre-hospital triage works on 4 codes as described in Table 1. However, since 2019, Emergency departments codify their triage on a 5-code basis. The two codes are not equivalent, as one is a standard for pre-hospital triage, while the other is a standard for hospital triage. Did the sardinian system differ from the national standards at the time of the study? The Authors should increase clarity on this point.
3) Thank you for this comment. We specified in the paper (Section 2.1) that although the five emergency department triage codes have been introduced in Italy since 2019, in Sardinia, the implementation of the national triage guidelines took place in March 2023, while the implementation phases of the five new codes only started later.
4) The study only included patients who were willing to participate. It is unclear if the resulting sample is representative of the whole patients population (e.g., age distribution, gender ratio, socioeconomic factors, clinical condition, clinical relevance). This could lead to selection bias, limiting external validity. Moreover, the sample may not be representative of the general population, again limiting the study validity. Formal statistical tests should be performed to verify representativeness (es. goodness-of-fit test comparing sample demographics to patients demographics, and also to census data). If the sample is not representative, adjustments should be made (es. weighting factors) to correct for demographic imbalances.
4) Thank you for this comment. We are aware of the concern about the external validity of the study. We cannot prove that our sample is representative of the general population (we do not have access to this data), although we collected data in a small area of the region. We integrated and expanded this issue in the limitations section (third limitation, page 11).
5) The study mentions usage of two different quesitonnaires, one for patients, one for healthcare professionals. However, there is no real information on those questionnaires. What is their origin? were they validated? what are their reliability metrics?
5) We apologize for the inconvenience. The term questionnaire creates misunderstandings. It is more correct to talking about of check list with dichotomous answers and a choice of response options. In the manuscript, we better described the instrument by defining that as a check list (Section 2.3).
6) "The study also enrolled all patients who were transported by ambulance between 110
January and March 2020 ". Does "also" means there is another cohort enrolled in a different period?
6) We apologize for the syntax error in the sentence. The study includes only one cohort of patients. Now, we have deleted “also” (Page 3, section 2.1).
Results
7) The results section may need rework, as per eventual methods adjustments suggested. In particular, the statistical analysis seems mainly focused on descriptive aspects, and not accounting for potential sampling imbalances. Inferential elements are reduced to a single chi-squared test.
7) Thank you for your comment. Based on your comment 4, we highlighted the sample imbalances issue in the limitation section by reporting that: “Data collection was interrupted due to the SARS-CoV-2 pandemic, which resulted in a reduced sample size and a shorter than expected data collection period. This could have led to potential imbalances in the sample compared to the general population of ambulance service users. So, there may have been over- or under-representation of certain age groups, education levels or other demographic variables. These imbalances may have influenced the results and in particular the perceptions of appropriateness of ambulance use.”
8) No detail is provided to questionnaire items response.
8) Thank you for pointing out the need for more detail. In revising the manuscript, we expanded Methods (2.3 Instruments) section to include specific information on the instruments. These are checklists, not questionnaires with Likert scales, so having now changed this part, the results should be congruent. Also, in the results section we specified that forty-eight patients were excluded because they did not meet the inclusion criteria (Page 5).
Discussion
9) Results should be addressed in the context of current literature, instad of just summarizing the findings.
9) Thank you, we integrated the discussion by contextualising the results by referring to the literature (Pages 9-10).
10) Methodological limitation of the study are not taken into account in discussions. There is no deep evaluaiton of such limitation, nor alternative interpretations of the findings are provided.
10) Thank you for your comment. We addressed the issue of methodological limitations by integrating them in the appropriate section as suggested (section 4.1). In addition, the discussion was deepened with further explanations of the results (Pages 9-10).
11) There is no indication of possible generalization of the findings.
11) Thank you for this comment. In line also with the indications of reviewer 1, We discussed this issue in the limitation section (Page 11, Section 4.1, third and fourth paragraph).
Conclusions
12) Authors should consider to add insights on clinical relevance and policy implications of their findings.
12) Thank you. We integrated the section with clinical relevance and policy implication (Page 12, section 5).

Round 2
Reviewer 1 Report
Comments and Suggestions for Authors
The authors have addressed all my comments.
Author Response
Dear Reviewer,
thank you very much for this positive feedback. We are happy to have met your expectations.
Best regards,
The authors
Reviewer 3 Report
Comments and Suggestions for Authors
EMS appropriateness and primary care access are a very sensitive and priority topic, especially now in Italy. Findigs from even small local studies can influence regional protocols and resource allocation, especially if such studies are then published on a very reputable journal. To ensure that such study conclusions may be useful in both academic and policy-making setting, evidence‐based criteria for rigor and transparency should be accounted for. In the last revision, the paper clarity, especially the Introduction and Discussion sections, was improved. However, two critical areas remain unaddressed: (1) the instrument applied, and (2) the representativeness of the sample: a more extensive discussion of limitations cannot compensate for lack of thereof. 1) Instrument. Tools like questionnaires and checklists are used used to measure latent information; in this case, "perceived appropriateness". In order to ensure their measures and findigs are coherent and generalisable, their psychometric properties (i.e. Validity and Reliability) must be investigated. For newly developed tool, both such characteristics and the developement process should be detailed, so that any potential user can make its own informed decision on appropriateness and congruity of use. Cfr. https://www.sciencedirect.com/science/article/pii/S2590260124000110 This means that it is not enough to simply rebrand a questionnaire to checklist, if any psychometric characteristic is not investigated nor reported. 2) Sample. The acknowlegment that potential bias exists is not enough for a decision on admitting either its presence or its absence. A check for representativeness of recorded variables (i.e. age, gender, education level, familial network) should be reported. The Author's answer that "We cannot prove that our sample is representative of the general population (we do not have access to this data" cannot be correct, as this data was collected by the Authors, and data on population and families in the investigated period is available as Open Data from National Statistics Insitute (ISTAT 2021 census; cfr. https://www.istat.it/notizia/dati-per-sezioni-di-censimento/). In this sense, Authors should also adjust their results depending on sample imbalances, applying appropriate weighting or stratification in their analyses, or at minimum conducting a sensitivity analysis to show how results change under different weighting schemes. The manuscript is not fundamentally flawed, and as such I would reccomend a Major Revision. If the mentioned issues are not resolved upon resubmission, I'd recommend rejection.
Author Response
Reviewer 3
Legend: Comment, Answer
Comment:
EMS appropriateness and primary care access are a very sensitive and priority topic, especially now in Italy. Findigs from even small local studies can influence regional protocols and resource allocation, especially if such studies are then published on a very reputable journal. To ensure that such study conclusions may be useful in both academic and policy-making setting, evidence‐based criteria for rigor and transparency should be accounted for. In the last revision, the paper clarity, especially the Introduction and Discussion sections, was improved. However, two critical areas remain unaddressed: (1) the instrument applied, and (2) the representativeness of the sample: a more extensive discussion of limitations cannot compensate for lack of thereof.
1) Instrument. Tools like questionnaires and checklists are used used to measure latent information; in this case, "perceived appropriateness". In order to ensure their measures and findigs are coherent and generalisable, their psychometric properties (i.e. Validity and Reliability) must be investigated. For newly developed tool, both such characteristics and the developement process should be detailed, so that any potential user can make its own informed decision on appropriateness and congruity of use. Cfr. https://www.sciencedirect.com/science/article/pii/S2590260124000110 This means that it is not enough to simply rebrand a questionnaire to checklist, if any psychometric characteristic is not investigated nor reported.
Answer:
Thank you for pointing this out. We agree that both checklists and questionnaires are used to measure latent variables. The authors for this study used instruments from the reference literature (Richards JR, Ferrall SJ. Inappropriate Use of Emergency Medical Services Transport: Comparison of Provider and Patient Perspectives. Acad Emerg Med. 1999;6(1):14-20. doi:https://doi.org/10.1111/j.1553-2712.1999.tb00088.x) and adapted them to the context. The instrument aims to collect descriptive and factual data on patients and the circumstances of the emergency call. The questions include demographic information (gender, age, educational qualification) and factual circumstances (e.g., perceived need for the ambulance, reasons for the call, time elapsed before the call). The instrument does not contain psychometric scales that measure latent constructs or complex attitudes. The instrument does not contain multi-item scales that would require internal consistency ratings (e.g., Cronbach's Alpha). Each question collects a specific and distinct piece of information, and there is no underlying construct that is measured by multiple items. In this sense, “perceived appropriateness” is not a latent variable, but is represented by a dichotomously answered question. For example:
1) Did you consider it necessary to activate an ambulance? ?
YES ? NO
2) If "YES" why? ?
-Inability to reach the emergency room due to episode complications (e.g., pain, dizziness, etc.) ?
-Emergency room too far away ?
-Lack of transportation ?
-Lack of network (friends and/or relatives with means of transportation) ?
-Other (specify)_______________________________________________________
3) If "NO" why? ?
-I was persuaded by friends and/or family ? The ambulance was called by a third party ?
-Other (specify)_______________________________________________________
In fact, the analyses performed for this study are primarily descriptive (frequencies, percentages) and analyses of comparisons between categorical variables (contingency tables, chi-square). The focus of these analyses is on the accuracy and interpretation of individual responses, rather than construct validity and internal consistency of the scales.
Comment:
2) Sample. The acknowlegment that potential bias exists is not enough for a decision on admitting either its presence or its absence. A check for representativeness of recorded variables (i.e. age, gender, education level, familial network) should be reported. The Author's answer that "We cannot prove that our sample is representative of the general population (we do not have access to this data" cannot be correct, as this data was collected by the Authors, and data on population and families in the investigated period is available as Open Data from National Statistics Insitute (ISTAT 2021 census; cfr. https://www.istat.it/notizia/dati-per-sezioni-di-censimento/). In this sense, Authors should also adjust their results depending on sample imbalances, applying appropriate weighting or stratification in their analyses, or at minimum conducting a sensitivity analysis to show how results change under different weighting schemes. The manuscript is not fundamentally flawed, and as such I would reccomend a Major Revision. If the mentioned issues are not resolved upon resubmission, I'd recommend rejection.
Answer:
Thank you for this comment. We recognize the importance of the sample representativeness as a crucial issue.
We followed the reviewer's suggestion assessing any imbalances and then applied weighting. Unfortunately, the age ranges of our sample did not match those given by the census for the general population, so they were not comparable. Therefore, we decided to weight by level of education. As such, we re-carried out all analyses and rewrote the results. All the change are highlighted in yellow in the manuscript.
We hope that our revisions will meet the reviewer's expectations and that our work will be positively evaluated.